# Characterization and Differentiation of Grain Proteomes from Wild-Type Puroindoline and Variants in Wheat

**DOI:** 10.3390/plants12101979

**Published:** 2023-05-15

**Authors:** Peixun Liu, Zehou Liu, Xiaofei Ma, Hongshen Wan, Jianmin Zheng, Jiangtao Luo, Qingyan Deng, Qiang Mao, Xiaoye Li, Zongjun Pu

**Affiliations:** 1Crop Research Institute, Sichuan Academy of Agricultural Sciences, Environmentally Friendly Crop Germplasm Innovation and Genetic Improvement Key Laboratory of Sichuan Province, Key Laboratory of Wheat Biology and Genetic Improvement on Southwestern China, Ministry of Agriculture and Rural Areas, Chengdu 610066, China; littlefarmer@163.com (P.L.);; 2Wheat Research Institute, Shanxi Agricultural University, Linfen 041000, China

**Keywords:** *Triticum aestivum*, grain hardness, proteomics, puroindoline genotypes

## Abstract

Premium wheat with a high end-use quality is generally lacking in China, especially high-quality hard and soft wheat. *Pina-D1* and *Pinb-D1* (*puroindoline* genes) influence wheat grain hardness (i.e., important wheat quality-related parameter) and are among the main targets in wheat breeding programs. However, the mechanism by which puroindoline genes control grain hardness remains unclear. In this study, three hard wheat puroindoline variants (MY26, GX3, and ZM1) were compared with a soft wheat variety (CM605) containing the wild-type puroindoline genotype. Specifically, proteomic methods were used to screen for differentially abundant proteins (DAPs). In total, 6253 proteins were identified and quantified via a high-throughput tandem mass tag quantitative proteomic analysis. Of the 208 DAPs, 115, 116, and 99 proteins were differentially expressed between MY26, GX3, and ZM1 (hard wheat varieties) and CM605, respectively. The cluster analysis of protein relative abundances divided the proteins into six clusters. Of these proteins, 67 and 41 proteins were, respectively, more and less abundant in CM605 than in MY26, GX3, and ZM1. Enrichment analyses detected six GO terms, five KEGG pathways, and five IPR terms that were shared by all three comparisons. Furthermore, 12 proteins associated with these terms or pathways were found to be differentially expressed in each comparison. These proteins, which included cysteine proteinase inhibitors, invertases, low-molecular-weight glutenin subunits, and alpha amylase inhibitors, may be involved in the regulation of grain hardness. The candidate genes identified in this study may be relevant for future analyses of the regulatory mechanism underlying grain hardness.

## 1. Introduction

Hexaploid bread wheat (*Triticum aestivum* L.) is a critical food crop worldwide. In China, which is the largest producer of bread wheat, the wheat-growing region comprises approximately 24 million hectares and includes basins, hills, plains, plateaus, and mountains. New high-quality and genetically diverse wheat varieties are urgently needed to meet the increasing demand for wheat due to societal and economic development and increases in living standards. Currently, wheat varieties in China mainly consist of general and mixed wheat, which are insufficient for satisfying the processing demands in the wheat industry. Specifically, there is a lack of premium quality wheat for special end-uses, especially hard and soft wheat types [1].

Grain hardness is an important wheat quality-related trait and one of the main phenotypic targets among wheat breeders [2]. Most wheat varieties are divided into two main classes on the basis of the kernel texture (i.e., hard and soft), with the remaining varieties classified as either medium-hard or medium-soft wheat. Hard and soft wheat varieties mill differently and have quantitative differences (e.g., break flour, total flour, starch damage, and flour particle size). Compared with soft wheat flour, hard wheat flour consists of larger particles and more damaged starch, leading to greater water absorption during dough formation [3]. Hence, hard and soft wheat grains have different commercial end-uses. Hard wheat grains are primarily used for producing bread, whereas soft wheat grains are used for the production of cookies, cakes, and confectionary products [4]. Therefore, clarifying the genetic basis of wheat grain hardness is critical for increasing wheat quality as well as for breeding varieties to address various consumer needs [5].

Wheat grain hardness is highly correlated with the friabilin protein, which is abundant in soft wheat but rare or absent in hard wheat. Friabilin consists of two proteins, puroindoline a and b, which are encoded by *Pina-D1* and *Pinb-D1*, respectively [6]. Both genes (447 bases and no introns) are located on the short arm of chromosome 5D and are closely linked. Because they encode proteins that combine to form a complex, they represent the key genes associated with wheat grain hardness. The Pina and Pinb proteins are rich in tryptophan, contain the AAI_SS domain specific to higher plants, and consist of 148 amino acids, with a molecular weight of approximately 13 kDa. The polymorphism of the *Pin* genes explains more than 60% of the diversity in kernel hardness [7]. Wheat grains are soft when both *Pin* genes are in their “wild state” *(Pina-D1a* and *Pinb-D1a*), whereas if one of these genes is absent or mutated, wheat grains will have a hard texture [8].

When the wild-type *Pinb-D1* allele is transferred into hard wheat plants, the transgenic wheat grains are reportedly soft, and there is a substantial decrease in kernel hardness [9]. An earlier analysis of transgenic durum wheat grains indicated that *Pina* overexpression results in decreased grain hardness [10]. In another previous study, the translocation of wild-type puroindoline-encoding genes into durum wheat resulted in the formation of soft grains [11]. The expression of *Pina-D1* in transgenic durum wheat lines leads to the production of fine flour particles and decreased starch damage [12].

The considerable diversity in the *Pina-D1* and *Pinb-D1* alleles affects the wheat grain texture. To date, 27 *Pina* alleles (*Pina-D1a-y*, *v2*, and *w2*) and 30 *Pinb* alleles (*Pinb-D1a-w* and *aa-ag*) have been identified in common wheat and other related diploid and hexaploid species [13]. Soft wheat needs both functional Pina and Pinb proteins or the wild-type alleles (*Pina-D1a* and *Pinb-D1a*) of both genes.

We previously examined the grain hardness and the genotypes at the puroindoline gene locus of wheat varieties collected from each wheat ecological region in China. Three variants of *Pinb-D1* were detected, of which *Pinb-D1b* was the most common genotype. Moreover, it was detected in Mianyang 26 (MY26), in which puroindoline b contains a glycine-to-serine sequence change because of a single nucleotide mutation (G223A) [14]. Additionally, *Pinb-D1c*, which was identified in Guixie 3 (GX3), has a single nucleotide mutation (T266C) that leads to a leucine-to-proline change at position 60 [15]. The deletion of one nucleotide (A213) in the *Pinb-D1p* allele in Zhemai 1 (ZM1) results in a lack of Pinb protein [16]. In the current study, the proteomes of three hard wheat varieties (MY26, GX3, and ZM1) with diverse puroindoline-encoding genes were compared with that of the soft wheat variety Chuanmai 605 (CM605), which contains wild-type puroindoline-encoding alleles, to screen for differentially abundant proteins (DAPs). The study data provide the foundation for future investigations on the molecular mechanism underlying wheat grain hardness.

## 2. Results and Discussion

### 2.1. Examination of the Puroindoline Genotypes and Grain Hardness Indices of the Experimental Materials

To analyze the DAPs in wheat grains with varying hardness indices, four varieties with diverse puroindoline genotypes (CM605, ZM1, MY26, and GX3) were selected for this study. The soft wheat variety CM605, which carries the wild-type puroindoline-encoding alleles (*Pina-Dla*/*Pinb-D1a*), had a grain hardness index of 33.1. In contrast, the hard wheat varieties MY26, GX3, and ZM1 had grain hardness indices of 65.8, 63.0, and 60.7 and *Pina-Dla*/*Pinb-D1b*, *Pina-Dla*/*Pinb-Dlc*, and *Pina-Dla*/*Pinb-D1p* genotypes, respectively (Table 1).

### 2.2. Protein Identification and Quantification

A total of 66,369 matched spectra, 32,547 peptides, and 6253 proteins were identified by the tandem mass tag (TMT) analysis of the four wheat varieties. Relative quantitative data were obtained for the 6253 identified proteins. Additionally, 2601 of the detected proteins were annotated according to all four of the following databases: Gene Ontology (GO), Kyoto Encyclopedia of Genes and Genomes (KEGG), Clusters of Orthologous Groups (COG), and InterPro (IPR) (Figure 1). The following criteria were used to identify significant DAPs: fold-change >2 (increased abundance) or <0.5 (decreased abundance) and a false discovery rate <0.05. Of the 6253 identified proteins, 208 were identified as DAPs. Moreover, 115 proteins were differentially expressed between MY26 and CM605, of which 37 and 78 proteins were significantly more and less abundant, respectively, in MY26 than in CM605. Furthermore, 116 proteins were differentially expressed between GX3 and CM605, of which 43 and 73 proteins were significantly more and less abundant, respectively, in GX3 than in CM605. Among the 99 proteins that were differentially expressed between ZM1 and CM605, 42 and 57 were significantly more and less abundant, respectively, in ZM1 than in CM605.

The techniques used in this study enabled a high-throughput and high-resolution proteomic analysis. In earlier studies, 1211 quinoa proteins were identified using a label-free quantification method [17]; 6061 proteins in wheat grains were identified by a TMT analysis [18], and 6958 wheat proteins were identified on the basis of iTRAQ data [19]. More specifically, TMT analyses are performed using a multiplexed protein identification and quantitation strategy involving isotope-labeling techniques that provide relative and absolute protein quantities in complex mixtures [20]. In the current study, 6253 proteins were identified and quantified in the grains of four wheat varieties.

### 2.3. Cluster Analysis of Protein Relative Abundances

A cluster analysis of protein relative abundances was completed to determine the correlation between protein relative abundances and puroindoline genotypes. The protein relative abundance for each sample was obtained. The expression data for all samples were combined for the C-means cluster analysis. The results of the cluster analysis are presented in Figure 2. Proteins were classified into six clusters according to their expression levels. Sixty-seven proteins were significantly more abundant in CM605 than in MY26, GX3, and ZM1 and were classified in Cluster 4. In contrast, 41 proteins were significantly less abundant in CM605 than in MY26, GX3, and ZM1 and were classified in Cluster 5 (Table 2).

### 2.4. GO Enrichment Analysis of DAPs

Differentially abundant proteins detected by the comparisons between the wheat varieties with differing puroindoline genotypes and the wheat variety with the wild-type puroindoline genotype were included in the GO enrichment analysis. For the MY26 vs. CM605 comparison, the GO enrichment analysis assigned 69 GO terms to 115 DAPs. Among these GO terms, 21 were significantly enriched (*p* < 0.05). Notably, some proteins were annotated with multiple GO terms. For the GX3 vs. CM605 comparison, 116 DAPs were annotated with 67 GO terms, of which 17 were significantly enriched. For the ZM1 vs. CM605 comparison, 99 DAPs were annotated with 45 GO terms, among which 16 were significantly enriched. Of the enriched GO terms assigned to the DAPs, the following six were shared by all three comparisons: chitin catabolic process, cell wall macromolecule catabolic process, and response to stress (biological process terms) and enzyme inhibitor activity, chitin binding, and chitinase activity (molecular function terms) (Figure 3).

The comparisons of the wheat varieties detected six proteins that were annotated with the above-mentioned six common GO terms (Table 3). Four chitinase-related GO terms (chitin catabolic process, cell wall macromolecule catabolic process, chitin binding, and chitinase activity) were assigned to TraesCS1D01G249600.1. One peroxidase-related term (response to stress) was assigned to two DAPs (TraesCS3B01G577900.1 and TraesCS3B01G578000.1). Another enriched GO term in all three comparisons (enzyme inhibitor activity, which is associated with cysteine proteinase inhibitors and invertase inhibitors) was assigned to a proteinase inhibitor protein (TraesCS4A01G052100.1) and two invertase inhibitor proteins (TraesCS4A01G459900.1 and TraesCS4A01G460900.1).

Cysteine proteinases exist in a wide variety of plants and are involved in several physiological processes. Most phytocystatins are inhibitors of cysteine proteases and have multiple important functions in plants. For example, they control various physiological and cellular processes in plants, while also inhibiting the activities of exogenous cysteine proteases that are secreted by herbivorous arthropods and pathogens to digest or colonize plant tissues [21,22]. Earlier research established clear correlations among storage protein deposition, cystatin biosynthesis, and decreased cysteine protease activities in storage organs. The functional relationship between cystatins and cathepsin L-like proteases was previously inferred on the basis of their involvement in the mobilization of storage proteins during the germination of barley seeds [23]. A cysteine proteinase (gliadian) that is secreted into the endosperm to digest storage proteins is reportedly regulated by intrinsic cystatins in wheat [24]. Another study identified two wheat cystatins (WC1 and WC4) with inhibitory effects on hydrolysis [25]. In barley, the downregulated production of a cystatin (HvIcy-2), which is one of the proteinaceous inhibitors of the cathepsin F-like protease, influences the grain-filling process [26]. Accordingly, cysteine proteinase inhibitors might contribute to the regulation of grain hardness by affecting the synthesis or hydrolysis of grain storage proteins.

Invertases are hydrolases that catalyze a reaction that converts sucrose to glucose and fructose. These enzymes are widely found in plants, animals, and microorganisms. On the basis of their solubility, localization, and pH optima, the invertases in higher plants can be divided into the following three groups: cytoplasmic, vacuolar, and cell wall invertases [27]. The unique expression pattern of the rice *GIF1* gene, which encodes a cell wall invertase, reflects the close relationship between cell wall invertases and the kernel weight [28]. A previous study on maize showed that the constitutive expression of a cell wall invertase-encoding gene increases the total starch content by up to 20% in transgenic plants (relative to the corresponding content in wild-type control plants) [29]. Plastidic invertases, which are responsible for all of the invertase activities in the chloroplasts of *Arabidopsis thaliana* leaves, are required for starch accumulation [30]. Some invertases can modulate the starch content in plants, thereby indirectly affecting grain hardness. However, the specific relationship between invertase functions and grain hardness remains undetermined.

### 2.5. KEGG Pathway Enrichment Analysis of DAPs

The DAPs detected by the three comparisons also underwent a KEGG pathway enrichment analysis. The 20 most enriched KEGG pathways among the DAPs revealed by the three comparisons are presented in Figure 4. Of these enriched KEGG pathways, the following five were common to the three comparisons: ‘glycosphingolipid biosynthesis–globo and isoglobo series’, ‘sphingolipid metabolism’, ‘fluid shear stress and atherosclerosis’, ‘MAPK signaling pathway–plant’, and ‘amino sugar and nucleotide sugar metabolism’. The following two DAPs were associated with four enriched KEGG pathways: TraesCS1D01G249600.1 (chitinase) and TraesCS5B01G011700.1 (alpha-galactosidase; α-Gal) (Table 3).

Alpha-galactosidase (EC 3.2.1.22) is a type of exoglycosidase that can specifically catalyze the hydrolysis of α-galactosidic bonds. It has been detected in animals, plants, and microorganisms (archaea, bacteria, and fungi). However, compared with the research on α-Gal in microorganisms, there have been relatively few investigations on α-Gal in plants. Nevertheless, previous research demonstrated that α-Gal in plants is often involved in important physiological processes, including leaf development and senescence [31], seed development and germination [32], fruit softening and ripening [33], and stress responses [34]. Unfortunately, the effects of α-Gal on grain hardness are unknown.

### 2.6. IPR Enrichment Analysis of DAPs

The enriched IPR terms among the DAPs detected by the three comparisons were also determined. The 10 most enriched IPR terms among the DAPs are provided in Figure 5. The following five IPR terms were common to the three comparisons: ‘protein of unknown function DUF538’, ‘chitin-binding, type 1’, ‘glycoside hydrolase, family 19, catalytic’, ‘pectinesterase inhibitor’, and ‘bifunctional inhibitor/plant lipid transfer protein/seed storage helical domain’. The ‘protein of unknown function DUF538′ IPR term was assigned to two proteins (TraesCS5B01G267400.1 and TraesCSU01G074400.1), both of which were annotated as a plant/protein (protein of unknown function). Both ‘chitin-binding, type 1′ and ‘glycoside hydrolase, family 19, catalytic’ were assigned to TraesCS1D01G249600.1, which was annotated as a chitinase. The ‘pectinesterase inhibitor’ term was assigned to two proteins (TraesCS4A01G459900.1 and TraesCS4A01G460900.1), which were annotated as invertase inhibitors. The ‘bifunctional inhibitor/plant lipid transfer protein/seed storage helical domain’ term was assigned to three proteins, of which two (TraesCS1B01G011600.1 and TraesCS1B01G011700.1) were annotated as low-molecular-weight glutenin subunits (LMW-GSs) and one (TraesCS2B01G004800.1) was annotated as an alpha amylase inhibitor.

Low-molecular-weight glutenin subunits are polymeric protein components in the wheat endosperm. Their ability to form inter-molecular disulfide bonds with each other and/or with high-molecular-weight glutenin subunits is important for the formation of glutenin polymers and determines the processing properties of wheat dough [35]. A single wheat variety may contain 7–16 different LMW-GSs [36]. Moreover, each LMW-GS differentially influences the processing quality of flour [37]. Generally, most subunits (e.g., Glu-A3d, Glu-B3d, and Glu-D3d) positively affect dough strength. However, other subunits (e.g., Glu-B3j) are negatively correlated with the rheological properties of dough [38]. In the present study, the relative expression level of two LMW-GSs (TraesCS1B01G011600.1 and TraesCS1B01G011700.1) had a negative correlation with the wheat grain hardness index, suggesting they may have important functions affecting wheat grain hardness.

The grain starch content is reportedly negatively correlated with grain hardness, with increases in the starch content potentially resulting in the production of grains with a relatively soft endosperm texture [39]. In the current study, TraesCS2B01G004800.1 was annotated as an alpha amylase inhibitor that might restrict the hydrolysis of starch, ultimately leading to an increase in the total starch content of wheat grains. The Pfam database contains a large collection of multiple sequence alignments and hidden Markov models for many common protein families [40]. We determined that the Pfam ID (PF00234: protease inhibitor/seed storage/LTP family) of TraesCS2B01G004800.1 is the same as that of puroindoline a (TraesCS5D01G004100.1) and puroindoline b (TraesCS5D01G004300.1), with the latter protein identified as the main determinant of wheat grain hardness [7]. Accordingly, our findings are suggestive of a potentially critical relationship between TraesCS2B01G004800.1 and grain hardness.

### 2.7. Terms/Pathways/Proteins Common to All Three Comparisons

By analyzing the proteins annotated with the six GO terms and five IPR terms or assigned to the five KEGG pathways that were enriched in all three comparisons, 12 proteins annotated with these terms or assigned to these pathways were revealed to be differentially expressed in each comparison (Table 3). Of these 12 proteins, only two were upregulated in the wheat varieties with variant puroindoline genotypes (compared with the wheat variety with the wild-type puroindoline genotype); both proteins belonged to Cluster 5. The remaining 10 DAPs were downregulated in the wheat varieties with variant puroindoline genotypes and belonged to Cluster 4 (Figure 2). The functions of these proteins and their effects on wheat grain hardness are described above. According to our results, several DAPs identified as cysteine proteinase inhibitors, invertases, LMW-GSs, and alpha amylase inhibitors may have regulatory effects on wheat grain hardness. However, the potential relationships between these proteins and grain hardness will need to be experimentally verified.

## 3. Materials and Methods

### 3.1. Plant Materials

In a previous study, more than 100 wheat varieties were collected from each wheat ecological region in China, after which the puroindoline gene-encoding locus was genotyped and the grain hardness index was calculated. Four wheat varieties that differed in terms of their puroindoline genotypes and grain hardness indices were selected for this study. More specifically, CM605, MY26, GX3, and ZM1 had puroindoline genotypes of *Pina-Dla*/*Pinb-D1a*, *Pina-Dla*/*Pinb-D1b*, *Pina-Dla*/*Pinb-Dlc*, and *Pina-Dla*/*Pinb-D1p*, respectively. Thus, these four varieties were classified into the following two categories: wild-type puroindoline genotype (*Pina-Dla*/*Pinb-D1a*) with a soft grain texture and variants (*Pina-Dla*/*Pinb-D1b*, *Pina-Dla*/*Pinb-Dlc*, and *Pina-Dla*/*Pinb-D1p*) with a hard grain texture (Table 1).

The wheat cultivars were grown in an experimental field (36°14′ N, 111°58′ E) at The Wheat Research Institute, Shanxi Agricultural University (Linfen, Shanxi Province, China) from October 2019 to May 2020. For each wheat genotype, individual pods were considered as a biological replicate. Seeds were harvested from the naturally matured spikes, and the moisture content of grain was less than 12%. Three samples were collected per plot, and then each sample was examined three times. The grain hardness index was determined using approximately 100 g seeds per sample and the Single Kernel Characterization System (Model 4100; Perten Instruments, PerkinElmer, Waltham, MA, USA).

### 3.2. Protein Extraction

Individual samples were ground in liquid nitrogen. The ground material was resuspended in SDT lysis buffer (4% SDS, 100 mM DTT, and 10 mM TEAB) prior to a 5 min ultrasonication on ice. The lysate was incubated at 95 °C for 8 min and then centrifuged at 12,000× *g* for 15 min at 4 °C. The proteins in the supernatant were reduced with 10 mM DTT for 1 h at 56 °C and then alkylated with sufficient iodoacetamide for 1 h at room temperature in darkness. Precooled acetone (4-times volume) was added to the samples, which were then vortexed and incubated at −20 °C for at least 2 h. Samples were centrifuged at 12,000× *g* for 15 min at 4 °C, and the precipitate was collected. After washing with 1 mL cold acetone, the pellet was dissolved in dissolution buffer (8 M urea and 100 mM TEAB, pH 8.5). The protein concentration was determined on the basis of a Bradford protein assay. Next, 20 µg protein samples were analyzed by 12% SDS-PAGE initially at 80 V for 20 min and then at 120 V for 90 min. The gel was stained using Coomassie brilliant blue R-250 and destained until the bands were clear.

### 3.3. TMT Labeling of Peptides

Each protein sample was mixed with DB dissolution buffer (8 M urea and 100 mM TEAB, pH 8.5) for a total volume of 100 µL. Next, 1.5 µL trypsin and 100 mM TEAB buffer were added, and the samples were mixed and digested at 37 °C for 4 h, after which 1.5 µL trypsin and 2 µL CaCl_2_ (1 mol/L) were added to each sample before an overnight digestion. Formic acid was added to the digested sample, and the pH was adjusted (<3). The mixture was centrifuged at 12,000× *g* for 5 min at room temperature. The supernatant was slowly loaded onto a C18 desalting column, which was washed three times with washing buffer (0.1% formic acid and 3% acetonitrile) before samples were eluted using elution buffer (0.1% formic acid and 70% acetonitrile). The eluants were collected and lyophilized. Next, 100 µL 0.1 M TEAB buffer was added to reconstitute the samples, which were then mixed with 41 µL acetonitrile-dissolved TMT labeling reagent. The samples were shaken for 2 h at room temperature. The reaction was terminated by adding 8% ammonia. All labeled samples were mixed (equal volume), desalted, and lyophilized.

### 3.4. Separation of Fractions

Mobile phases A (2% acetonitrile; pH adjusted to 10.0 using ammonium hydroxide) and B (98% acetonitrile) were used for the gradient elution. The lyophilized powder was dissolved in solution A and centrifuged at 12,000× *g* for 10 min at room temperature. The sample was fractionated using a C18 column (Waters BEH C18, 4.6 × 250 mm, 5 µm) and a Rigol L3000 HPLC system. The column oven was set at 45 °C. The eluates were monitored at a UV wavelength of 214 nm. Fractions were collected at a rate of one tube per minute for a total of 10 fractions. All fractions were dried under vacuum conditions and reconstituted in 0.1% (*v*/*v*) formic acid in water.

### 3.5. LC-MS/MS Analysis

Shotgun proteomic analyses were performed using an EASY-nLC™ 1200 UHPLC system (Thermo Fisher, Waltham, MA, USA) coupled with a Q Exactive™ HF-X mass spectrometer (Thermo Fisher, Waltham, MA, USA) at Novogene Genetics, Beijing, China. Specifically, 1 µg sample was injected into a C18 Nano-Trap column (4.5 cm × 75 µm, 3 µm). Peptides were separated in an analytical column (15 cm × 150 µm, 1.9 µm) using a linear gradient elution. The separated peptides were analyzed using the Q Exactive™ HF-X mass spectrometer (Thermo Fisher, Waltham, MA, USA) combined with Nanospray Flex™ (electrospray ion source) (Thermo Fisher, Waltham, MA, USA), with a spray voltage of 2.3 kV and an ion transport capillary temperature of 320 °C. The full scan range was 350 to 1500 (*m*/*z*) with a resolution of 60,000 (at *m*/*z* 200). The automatic gain control target value was 3 × 10^6^, and the maximum ion injection time was 20 ms. The 40 most abundant precursors in the full scan were selected and fragmented by higher energy collisional dissociation for the MS/MS analysis with a 10-plex resolution of 45,000 (at *m*/*z* 200). The automatic gain control target value was 5 × 10^4^, and the maximum ion injection time was 86 ms. The normalized collision energy was set at 32%; the intensity threshold was 1.2 × 10^5^, and the dynamic exclusion parameter was 20 s.

### 3.6. Identification and Quantification of Proteins

The proteins corresponding to the spectra from each run were identified by screening the IWGSC RefSeq v1.0 annotated wheat genome database (https://wheat-urgi.versailles.inra.fr/ (accessed on 24 March 2017)) using the search engine Proteome Discoverer 2.4 (PD 2.4; Thermo). The search parameters were as follows: mass tolerance for the precursor ion, 10 ppm; mass tolerance for the product ion, 0.02 Da; fixed modification, carbamidomethylation; dynamic modifications, oxidation of methionine and TMT plex; and N-terminal modifications, acetylation, TMT plex, methionine loss, and methionine loss + acetylation. A maximum of two missed cleavage sites were allowed. To improve the quality of the analysis, PD 2.4 filtered the search results. Specifically, the peptide spectrum matches (PSMs) with a credibility score exceeding 99% were designated as credible PSMs. The identified proteins contained at least one unique peptide. The identified PSMs and proteins with a false discovery rate of no more than 1.0% were retained for further analyses. The protein relative abundances were analyzed by performing a T-test. Proteins with a relative expression level that differed significantly between the experimental and control groups (*p* < 0.05 and fold-change >2.00 or <0.50) were defined as DAPs.

### 3.7. Functional Characterization of DAPs

The GO and IPR functional analyses were conducted using the InterProScan program, and the results were compared with the information in non-redundant protein databases (Pfam, PRINTS, ProDom, SMART, ProSite, and PANTHER). The COG and KEGG databases were used to analyze the protein families and pathways. The DAPs were included in the volcano map analysis, cluster heat map analysis, and GO, IPR, and KEGG enrichment analyses.

## 4. Conclusions

To reveal the differences between soft wheat and hard wheat proteomes, three hard wheat varieties (MY26, GX3, and ZM1) with different puroindoline-encoding genes were compared with a soft wheat variety (CM605) with the wild-type puroindoline genotype. A total of 6253 proteins were identified and quantified. Furthermore, a cluster analysis of protein relative abundances detected 208 DAPs that were classified into six clusters. Among these DAPs, 67 and 41 proteins were significantly more and less abundant, respectively, in CM605 than in MY26, GX3, and ZM1. Moreover, six GO terms, five KEGG pathways, and five IPR terms were common among the three comparisons according to the enrichment analysis. Twelve proteins annotated with these terms or assigned to these pathways were differentially expressed in each group. Several proteins had been previously identified (e.g., cysteine proteinase inhibitor, invertase, LMW-GS, and alpha amylase inhibitor) and may be involved in the regulation of grain hardness. To the best of our knowledge, this is the first comparative proteomic analysis of hard and soft wheat varieties with differing puroindoline genotypes. The findings of this study lay the foundation for future investigations on the regulatory mechanism associated with puroindoline-encoding genes, while also providing researchers with potential candidate genes for future studies on wheat grain hardness.

## Figures and Tables

**Figure 1 plants-12-01979-f001:**
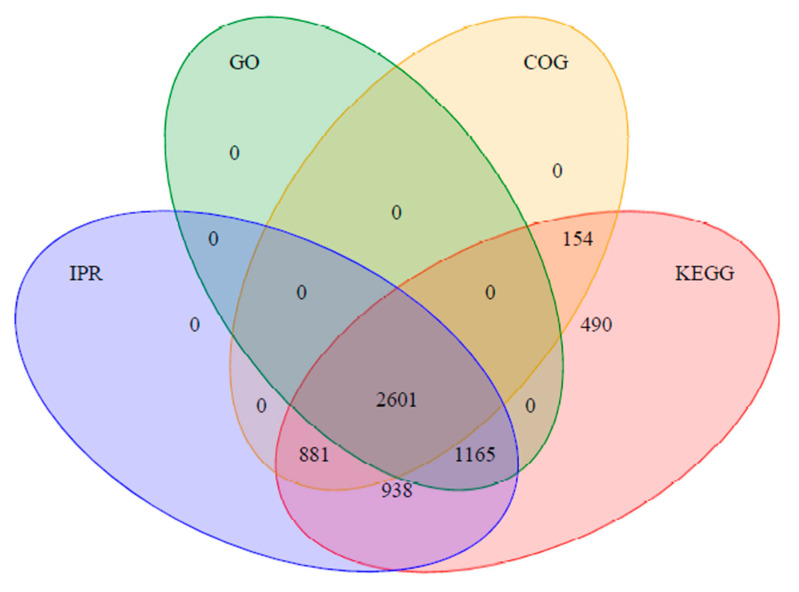
Venn diagram of the proteins identified according to four libraries.

**Figure 2 plants-12-01979-f002:**
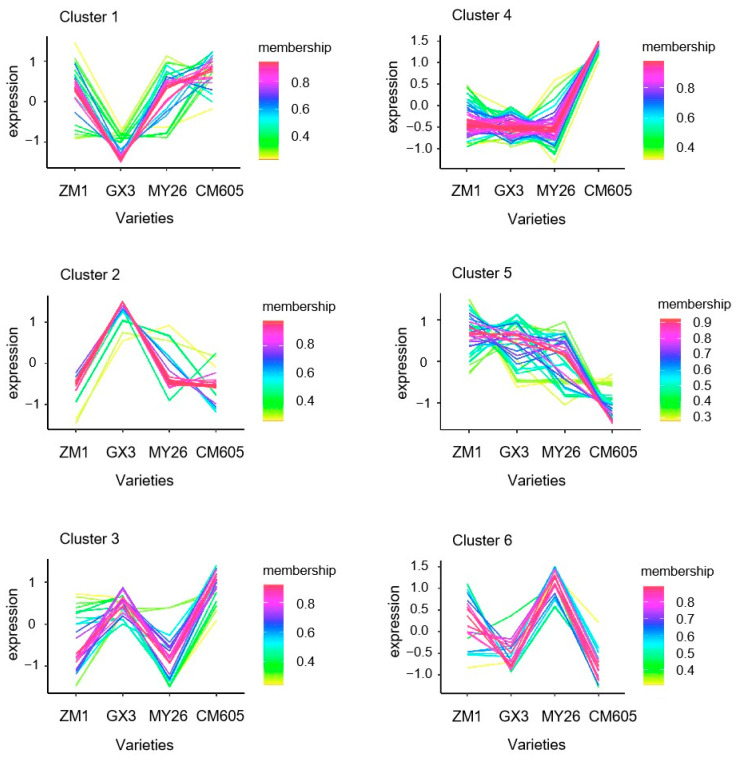
Results of the C-means cluster analysis of DAPs. Note: Varieties are presented on the abscissa, whereas the z-value corrected expression levels are presented on the ordinate (increasing values reflect increasing expression levels). Each line represents one protein. Differences in the membership values are indicated by color. A high membership value suggests the protein is close to the average in the category.

**Figure 3 plants-12-01979-f003:**
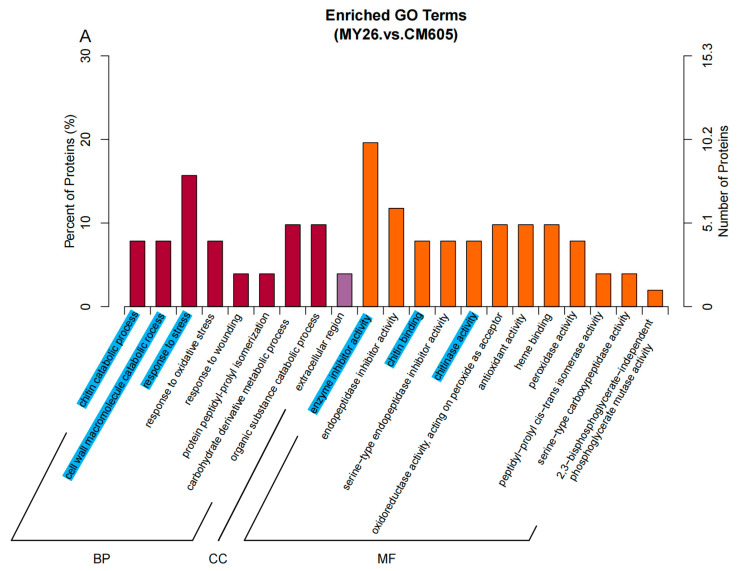
Enriched GO terms among the identified DAPs. Note: BP (biological process), CC (cellular component), and MF (molecular function).

**Figure 4 plants-12-01979-f004:**
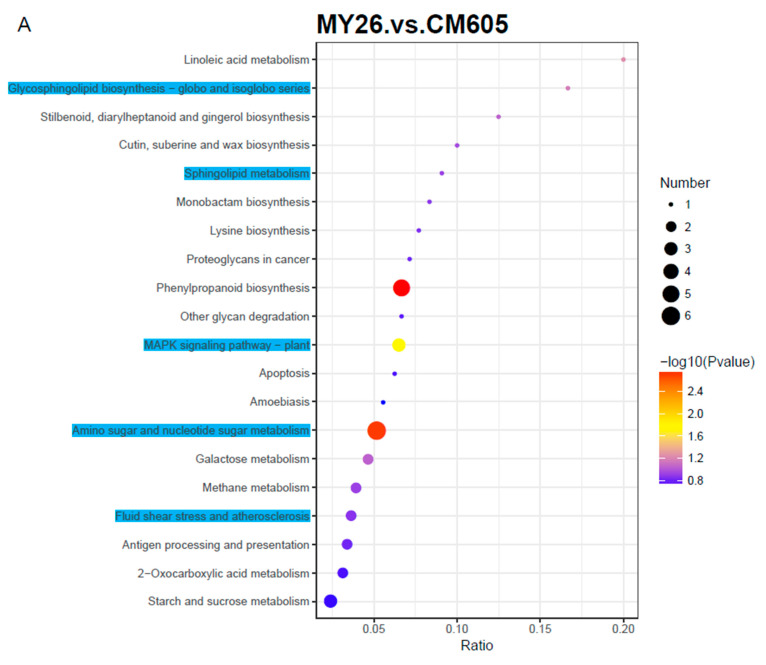
Top 20 enriched KEGG pathways among the DAPs. The solid black circles represent the number of DAPs, and the color represents −log10 (*p* value).

**Figure 5 plants-12-01979-f005:**
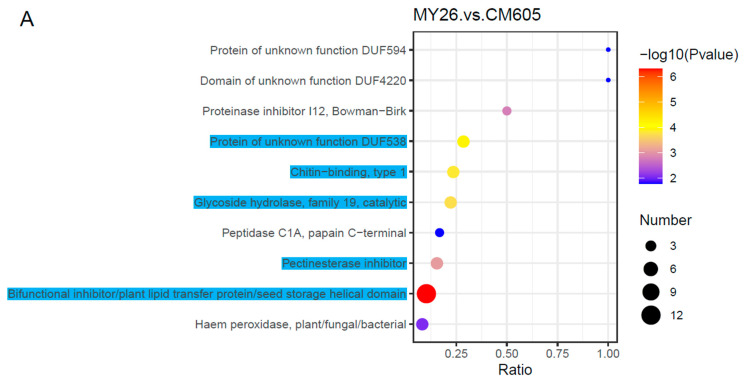
Top 10 enriched IPR terms among the DAPs. The solid black circles represent the number of DAPs, and the color represents −log10 (*p* value).

**Table 1 plants-12-01979-t001:** Names, genotypes, pedigree, and grain hardness indices of four wheat varieties.

No.	Variety	Puroindoline Genotype	Full Variety Name	Pedigree	Grain Hardness Index	Variety Origin
1	CM605	Pina-Dla/Pinb-D1a	Chuanmai 605	ICA80/99-6890//Chuanmai44	33.10 ± 1.05 ^dC^	Sichuan Province
2	MY26	Pina-Dla/Pinb-D1b	Mianyang 26	Mianyang20/Chuanyu9	65.80 ± 0.70 ^aA^	Sichuan Province
3	GX3	Pina-Dla/Pinb-Dlc	Guixie 3	*Triticum dicoccoides* (Israel)/*Avena fatua* L. *var. glabrata* Peterm.	63.00 ± 0.70 ^bB^	Guizhou Province
4	ZM1	Pina-Dla/Pinb-D1p	Zhemai 1	Linpuzao Landrace/Taiwa wheat (Japan)	60.70 ± 1.36 ^cB^	Zhejiang Province

In each column, different lowercase and uppercase letters indicate significant differences at the 0.05 and 0.01 levels, respectively.

**Table 2 plants-12-01979-t002:** The protein relative abundances between soft-grain and hard-grain wheat varieties (MY26 vs. CM605, GX3 vs. CM605, and ZM1 vs. CM605).

Protein/Gene	Description	The Average Number in MY26	The Average Number in GX3	The Average Number in ZM1	The Average Number in CM605	Fold Change (MY26 vs. CM605)	Fold Change (GX3 vs. CM605)	Fold Change (ZM1 vs. CM605)	Up/Down (vs. CM605)
TraesCS7B01G417700.1	Thaumatin-like protein	877.3	1019.9	1472.3	1796.4	0.5	0.6	0.8	Down
TraesCS1B01G011700.1	Low molecular weight glutenin subunit	2588.1	2597.1	2574.9	10737.1	0.2	0.2	0.2	Down
TraesCS3D01G013000.1	Sucrose phosphate synthase	435.1	365.1	739.1	918.6	0.5	0.4	0.8	Down
TraesCS1B01G104500.1	Carboxypeptidase	1077.9	1569.5	1099.0	2601.1	0.4	0.6	0.4	Down
TraesCS1B01G011600.1	Low molecular weight glutenin subunit	121.7	110.3	77.7	833.9	0.1	0.1	0.1	Down
TraesCS1A01G103600.1	Alpha-galactosidase	160.1	155.9	177.6	317.1	0.5	0.5	0.6	Down
TraesCS2A01G350700.1	Chitinase	715.2	953.6	777.4	1781.0	0.4	0.5	0.4	Down
TraesCS2B01G083800.1	Lactoylglutathione lyase	341.3	334.0	315.4	691.7	0.5	0.5	0.5	Down
TraesCS1B01G426100.1	Beta purothionin	1608.1	1394.2	3611.5	4821.7	0.3	0.3	0.7	Down
TraesCS3A01G087100.1	Cysteine synthase	38.2	38.3	39.4	101.3	0.4	0.4	0.4	Down
TraesCS5A01G095900.1	plant/protein (Protein of unknown function	221.3	178.7	335.6	766.4	0.3	0.2	0.4	Down
TraesCS5B01G011700.1	Alpha-galactosidase	179.0	192.1	180.8	461.1	0.4	0.4	0.4	Down
TraesCS7A01G404100.1	DUF1680 domain protein	674.3	508.0	652.7	1114.2	0.6	0.5	0.6	Down
TraesCS6B01G089500.2	ARC6	951.9	1103.7	1121.0	1921.2	0.5	0.6	0.6	Down
TraesCS1B01G081000.1	Protease inhibitor/seed storage/lipid transfer protein family protein	455.8	837.4	493.6	1745.1	0.3	0.5	0.3	Down
TraesCS5B01G424800.1	Protease inhibitor/seed storage/lipid transfer protein family protein	833.3	840.1	945.8	1892.0	0.4	0.4	0.5	Down
TraesCS5D01G275600.1	plant/protein (Protein of unknown function	283.9	329.6	359.2	568.9	0.5	0.6	0.6	Down
TraesCS2D01G582900.1	Peroxidase	738.3	968.2	904.5	1831.1	0.4	0.5	0.5	Down
TraesCS5B01G267400.1	plant/protein (Protein of unknown function	441.2	453.1	458.4	1134.2	0.4	0.4	0.4	Down
TraesCS7B01G128800.1	Epoxide hydrolase 2	2839.6	1595.8	1454.3	3298.6	0.9	0.5	0.4	Down
TraesCS2B01G004800.1	Alpha amylase inhibitor protein	3772.6	3459.6	2681.7	9006.9	0.4	0.4	0.3	Down
TraesCS5A01G267700.1	TraesCS5A01G267700.1	1251.1	1162.8	992.7	2304.2	0.5	0.5	0.4	Down
TraesCS1A01G423800.1	Late embryogenesis abundant protein Lea14	74.5	73.4	76.8	159.0	0.5	0.5	0.5	Down
TraesCS4A01G460900.1	Invertase inhibitor	544.1	994.0	695.7	2119.9	0.3	0.5	0.3	Down
TraesCS1A01G251900.1	Eukaryotic aspartyl protease family protein	85.2	115.3	110.7	215.1	0.4	0.5	0.5	Down
TraesCS1D01G249600.1	Chitinase	122.5	111.8	127.5	380.4	0.3	0.3	0.3	Down
TraesCSU01G202600.1	Kunitz trypsin inhibitor	1936.4	1995.3	3468.3	4822.2	0.4	0.4	0.7	Down
TraesCS3B01G360000.1	Protease inhibitor/seed storage/lipid transfer protein family protein	1509.9	1727.0	1458.7	3047.2	0.5	0.6	0.5	Down
TraesCS3A01G046000.1	Trypsin inhibitor	395.9	387.1	564.3	834.9	0.5	0.5	0.7	Down
TraesCS1A01G062700.1	Protease inhibitor/seed storage/lipid transfer family protein	34.9	69.0	29.8	186.9	0.2	0.4	0.2	Down
TraesCS6D01G280900.1	Lipoxygenase homology domain-containing protein 1	572.8	517.7	610.9	1080.0	0.5	0.5	0.6	Down
TraesCS4B01G333200.1	carboxyl-terminal peptidase (DUF239)	131.5	92.6	90.7	259.4	0.5	0.4	0.3	Down
TraesCS1B01G189500.1	Phenylalanine--tRNA ligase beta subunit	435.0	827.8	987.9	1201.1	0.4	0.7	0.8	Down
TraesCS1A01G028200.1	Chymotrypsin inhibitor	268.3	372.8	417.9	700.8	0.4	0.5	0.6	Down
TraesCS1A01G014100.1	Defensin	430.1	284.4	196.7	675.1	0.6	0.4	0.3	Down
TraesCS7D01G025500.1	Invertase/pectin methylesterase inhibitor family protein	328.8	246.4	286.6	615.0	0.5	0.4	0.5	Down
TraesCSU01G074400.1	plant/protein (Protein of unknown function	21.5	13.9	22.4	336.8	0.1	0.0	0.1	Down
TraesCS3A01G148700.1	Cold induced protein	670.2	836.4	1006.8	1501.9	0.4	0.6	0.7	Down
TraesCS1B01G018100.1	Defensin	105.9	107.5	85.0	242.0	0.4	0.4	0.4	Down
TraesCS4A01G075100.1	H-ATPase 9	171.7	224.3	197.8	388.7	0.4	0.6	0.5	Down
TraesCS1A01G366600.1	Invertase inhibitor	896.0	498.6	549.3	1131.8	0.8	0.4	0.5	Down
TraesCS2A01G519000.1	BTB/POZ domain-containing protein 10	104.7	98.9	105.6	313.4	0.3	0.3	0.3	Down
TraesCS4A01G459800.1	Plant invertase/pectin methylesterase inhibitor superfamily	44.4	45.6	45.1	157.5	0.3	0.3	0.3	Down
TraesCS7A01G203100.1	Caleosin	218.1	259.2	235.5	439.6	0.5	0.6	0.5	Down
TraesCS4A01G459900.1	Invertase inhibitor	865.7	940.7	614.7	2310.6	0.4	0.4	0.3	Down
TraesCS3A01G509000.1	Thioredoxin-like protein	232.7	304.7	259.1	537.0	0.4	0.6	0.5	Down
TraesCS4A01G052100.1	Cysteine proteinase inhibitor	653.1	622.5	712.4	2235.8	0.3	0.3	0.3	Down
TraesCS3B01G036400.1	Trypsin inhibitor	608.5	772.0	711.8	1227.1	0.5	0.6	0.6	Down
TraesCS4A01G067900.1	Glucan endo-1	54.5	46.6	70.6	101.4	0.5	0.5	0.7	Down
TraesCSU01G141300.1	TraesCSU01G141300.1	165.6	170.5	143.6	404.9	0.4	0.4	0.4	Down
TraesCS1A01G351300.1	Rapid alkalinization factor (RALF) family protein	348.2	322.0	402.9	676.5	0.5	0.5	0.6	Down
TraesCS1A01G308200.1	tRNA uridine 5-carboxymethylaminomethyl modification enzyme MnmG	265.4	237.9	288.2	764.3	0.3	0.3	0.4	Down
TraesCS5B01G016300.1	Thaumatin-like protein	108.5	148.4	169.8	273.9	0.4	0.5	0.6	Down
TraesCS6B01G418700.1	Chitinase	64.8	52.6	48.0	163.1	0.4	0.3	0.3	Down
TraesCS3B01G062700.1	Non-specific lipid-transfer protein	46.7	57.3	86.7	133.3	0.4	0.4	0.7	Down
TraesCS6A01G132200.1	Lipoxygenase	131.0	200.0	213.0	306.7	0.4	0.7	0.7	Down
TraesCS3B01G368200.1	Protein RETICULATA	348.5	186.7	168.8	463.3	0.8	0.4	0.4	Down
TraesCS2B01G501000.1	Non-specific lipid-transfer protein	93.4	97.8	142.3	216.1	0.4	0.5	0.7	Down
TraesCS1D01G233500.1	Late embryogenesis abundant (LEA) hydroxyproline-rich glycoprotein family	151.4	142.1	157.6	301.3	0.5	0.5	0.5	Down
TraesCS5A01G391500.1	Cysteine protease	147.8	183.7	130.2	321.5	0.5	0.6	0.4	Down
TraesCS4D01G241500.2	Caleosin	184.7	262.5	302.9	405.4	0.5	0.6	0.7	Down
TraesCS3A01G006000.1	Pectinesterase inhibitor	41.2	64.8	92.8	128.8	0.3	0.5	0.7	Down
TraesCS5D01G516500.1	RNA-binding protein	140.9	116.4	143.5	248.3	0.6	0.5	0.6	Down
TraesCS3D01G545600.1	TraesCS3D01G545600.1	522.5	344.5	257.7	967.8	0.5	0.4	0.3	Down
TraesCS6A01G041500.1	Transmembrane protein 97	45.6	55.4	50.3	125.3	0.4	0.4	0.4	Down
TraesCS2D01G440500.1	WD-40 repeat family protein-2	78.9	67.3	101.9	166.1	0.5	0.4	0.6	Down
TraesCS7A01G065700.1	Cysteine protease	139.1	145.9	168.6	336.4	0.4	0.4	0.5	Down
TraesCS4A01G418200.1	Starch synthase	7730.0	9118.1	6645.1	3331.8	2.3	2.7	2.0	Up
TraesCS4D01G098400.1	Protein disulfide-isomerase	1032.2	1179.2	917.8	546.2	1.9	2.2	1.7	Up
TraesCS3B01G423200.1	Fructose-bisphosphate aldolase	209.5	175.6	240.0	60.8	3.4	2.9	3.9	Up
TraesCS6B01G440200.1	Aconitate hydratase	1465.6	1512.4	1062.4	637.6	2.3	2.4	1.7	Up
TraesCS4D01G282400.1	Late embryogenesis abundant protein	2657.2	2871.0	3243.2	1072.1	2.5	2.7	3.0	Up
TraesCS1D01G000700.1	Gamma gliadin	521.2	469.2	399.0	250.9	2.1	1.9	1.6	Up
TraesCS3D01G114900.1	Heat-shock protein	866.8	1043.8	967.2	261.5	3.3	4.0	3.7	Up
TraesCS7D01G284600.1	Alpha-soluble NSF attachment protein	619.0	633.6	1459.8	580.5	1.1	1.1	2.5	Up
TraesCS3B01G578000.1	Peroxidase	194.4	235.2	234.3	64.5	3.0	3.6	3.6	Up
TraesCS7B01G116600.1	chr7B:135528384..135529775 (+ strand)	873.6	919.3	937.5	467.9	1.9	2.0	2.0	Up
TraesCS3B01G498300.2	Spermatogenesis-associated protein 20	79.7	310.7	292.3	74.5	1.1	4.2	3.9	Up
TraesCS1D01G087600.1	Carboxypeptidase	609.7	454.9	683.6	250.6	2.4	1.8	2.7	Up
TraesCS2D01G468200.1	Polyphenol oxidase	605.1	834.7	1043.1	412.1	1.5	2.0	2.5	Up
TraesCS5A01G481400.1	NAD(P)-binding rossmann-fold protein	80.8	82.0	80.3	36.3	2.2	2.3	2.2	Up
TraesCS1D01G009900.1	Low molecular weight glutenin subunit	808.8	738.5	993.6	473.7	1.7	1.6	2.1	Up
TraesCS6D01G005200.1	Protein disulfide-isomerase	322.2	533.9	444.7	222.6	1.4	2.4	2.0	Up
TraesCS4A01G103900.2	Glutathione-S-transferase	61.7	115.0	107.6	49.4	1.2	2.3	2.2	Up
TraesCS3D01G473000.1	S-formylglutathione hydrolase	97.6	97.3	86.1	47.4	2.1	2.1	1.8	Up
TraesCS3B01G577900.1	Peroxidase	452.7	554.2	569.6	124.4	3.6	4.5	4.6	Up
TraesCS4A01G092600.1	Heat-shock protein	79.8	84.6	647.1	89.9	0.9	0.9	7.2	Up
TraesCS1A01G007200.1	Gamma-gliadin	402.8	407.0	2447.5	346.5	1.2	1.2	7.1	Up
TraesCS6A01G049400.1	Alpha-gliadin	2311.8	3128.6	3396.3	1520.6	1.5	2.1	2.2	Up
TraesCS1D01G405600.1	Beta purothionin	1649.7	1496.6	1862.7	704.7	2.3	2.1	2.6	Up
TraesCS6B01G287800.1	Dimeric alpha-amylase inhibitor	46.9	72.3	79.3	38.2	1.2	1.9	2.1	Up
TraesCS7B01G434000.1	Purple acid phosphatase	1445.0	1441.0	4391.9	1420.7	1.0	1.0	3.1	Up
TraesCS2A01G371500.1	Short chain dehydrogenase/reductase	194.0	292.9	275.9	137.6	1.4	2.1	2.0	Up
TraesCS6B01G444700.1	Ubiquitin carboxyl-terminal hydrolase	152.5	204.4	151.8	80.7	1.9	2.5	1.9	Up
TraesCS1B01G445100.1	Oleosin	1038.5	1102.8	1005.8	417.8	2.5	2.6	2.4	Up
TraesCSU01G108700.1	Alpha gliadin	573.4	550.3	846.9	384.4	1.5	1.4	2.2	Up
TraesCS2B01G491400.1	Polyphenol oxidase	55.5	127.5	217.2	105.7	0.5	1.2	2.1	Up
TraesCS5D01G538800.1	Transferase	228.9	736.6	671.8	210.6	1.1	3.5	3.2	Up
TraesCS5A01G018900.1	Thaumatin-like protein	426.1	382.8	889.5	430.2	1.0	0.9	2.1	Up
TraesCS2B01G622400.1	Chitinase	565.4	514.7	878.1	357.9	1.6	1.4	2.5	Up
TraesCS4A01G304100.1	Histidine-containing phosphotransfer protein	614.2	815.5	646.8	388.9	1.6	2.1	1.7	Up
TraesCS6A01G049700.1	Alpha-gliadin	104.1	541.8	527.8	72.3	1.4	7.5	7.3	Up
TraesCS4A01G029900.1	Retrovirus-related Pol polyprotein from transposon TNT 1-94	328.3	292.5	329.7	119.7	2.7	2.4	2.8	Up
TraesCS3A01G025300.1	GDSL esterase/lipase	44.4	40.8	141.4	35.1	1.3	1.2	4.0	Up
TraesCS2A01G258700.1	transmembrane protein	827.3	758.9	748.0	242.0	3.4	3.1	3.1	Up
TraesCS6D01G007600.1	Protein ENHANCED DISEASE RESISTANCE 2-like	243.9	424.0	556.5	243.0	1.0	1.7	2.3	Up
TraesCS4D01G325500.1	Kinase	131.5	212.6	1878.9	161.0	0.8	1.3	11.7	Up
TraesCS5D01G105600.5	Collagen alpha-1(I) chain	364.9	402.3	270.3	171.0	2.1	2.4	1.6	Up

**Table 3 plants-12-01979-t003:** Results of the GO, KEGG, and IPR enrichment analyses of the DAPs revealed by three comparisons (MY26 vs. CM605, GX3 vs. CM605, and ZM1 vs. CM605).

No.	Protein ID	Description	GO Term	KEGG Pathway Map Title	IPR Title	Fold Change (MY26 vs. CM605)	Fold Change (GX3 vs. CM605)	Fold CHANGE (ZM1 vs. CM605)	Up/Down (vs. CM605)
1	TraesCS1B01G011600.1	Low molecular weight glutenin subunit	—	—	IPR016140 (Bifunctional inhibitor/plant lipid transfer protein/seed storage helical domain)	0.15	0.13	0.09	down
2	TraesCS1B01G011700.1	Low molecular weight glutenin subunit	—	—	IPR016140 (Bifunctional inhibitor/plant lipid transfer protein/seed storage helical domain)	0.24	0.24	0.24	down
3	TraesCS1D01G249600.1	Chitinase	GO:0008061 (chitin binding), GO:0004568 (chitinase activity), GO:0006032 (chitin catabolic process), GO:0016998 (cell wall macromolecule catabolic process)	map00520 (Amino sugar and nucleotide sugar metabolism), map04016 (MAPK signaling pathway—plant)	IPR001002 (Chitin-binding, type 1), IPR000726 (Glycoside hydrolase, family 19, catalytic)	0.32	0.29	0.34	down
4	TraesCS2B01G004800.1	Alpha amylase inhibitor protein	—	—	IPR016140 (Bifunctional inhibitor/plant lipid transfer protein/seed storage helical domain)	0.42	0.38	0.30	down
5	TraesCS3B01G577900.1	Peroxidase	GO:0006950 (response to stress)	—	—	3.64	4.45	4.58	up
6	TraesCS3B01G578000.1	Peroxidase	GO:0006950 (response to stress)	—	—	3.02	3.65	3.63	up
7	TraesCS4A01G052100.1	Cysteine proteinase inhibitor	GO:0004857 (enzyme inhibitor activity)	—	—	0.29	0.28	0.32	down
8	TraesCS4A01G459900.1	Invertase inhibitor	GO:0004857 (enzyme inhibitor activity)	—	IPR006501 (Pectinesterase inhibitor)	0.37	0.41	0.27	down
9	TraesCS4A01G460900.1	Invertase inhibitor	GO:0004857 (enzyme inhibitor activity)	—	IPR006501 (Pectinesterase inhibitor)	0.26	0.47	0.33	down
10	TraesCS5B01G011700.1	Alpha-galactosidase	—	map00603 (Glycosphingolipid biosynthesis—globo and isoglobo series), map00600 (Sphingolipid metabolism)	—	0.39	0.42	0.39	down
11	TraesCS5B01G267400.1	plant/protein (Protein of unknown function)	—	—	IPR007493 (Protein of unknown function DUF538)	0.39	0.40	0.40	down
12	TraesCSU01G074400.1	plant/protein (Protein of unknown function)	—	—	IPR007493 (Protein of unknown function DUF538)	0.06	0.04	0.07	down

## Data Availability

The mass spectrometry proteomics data have been deposited to the ProteomeXchange Consortium via the PRIDE partner repository with the dataset identifier PXD041989.

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
