# Peer review of "Characterization and Differentiation of Grain Proteomes from Wild-Type Puroindoline and Variants in Wheat"

_plants, 2023, doi:10.3390/plants12101979_

Round 1

Reviewer 1 Report

Wheat grain hardness is one of the main target of wheat breeding, proteomics were performed to elucidate the mechanism of grain hardness. However, despite the detailed explanation has been presented, several questions still need to be improved in the manuscript.

1> English should be improved.

2> The biological replicates of the measurement of grain hardness should be given and the data should be analyzed by one-way ANOVA. 

3> The differentially enriched proteins should be divided into upregulated and downregulated, the venn diagram of three group (MY26, GX3, and ZM1 VS CM605) should be displayed to figure out the DEPs shared in three groups and the specific DEPs in single group. 

4> "DEPs" should be "differentially enriched proteins ", not "differentially expressed proteins". 

5> The author mentioned that Chitinase and POD have no effect on grain hardness. Therefore, the section is meaningless and should be deleted.

6> The author failed to provide reasonable explanations of the formation mechnism of grain hardness,more details should be analyzed based on KEGG pathay.

Reviewer 2 Report

This manuscript is good and accepted with minor revision

Author Response

Response to Reviewer 2 Comments

Thanks for your comments. We have studied all of the comments carefully and made corrections in accordance with the reviewers’ suggestions. All the comments and questions are answered one by one, and the responses to all comments have been prepared and attached herewith below. We would like to express our gratitude to you again.

This manuscript is good and accepted with minor revision

Response: Thanks very much for your comments. 

Reviewer 3 Report

In this manuscript, three hard wheat purindoline variants (MY26, GX3, and ZM1) were compared with a soft wheat variety (CM605) containing a wild-type purindoline genotype using proteomic screening for different expression proteins (DEPs). High-throughput tandem mass tagging (TMT) quantitative proteomics was used to screen out the significantly expressed genes. It provides candidate genes for future analysis and lays a foundation for further revealing the regulatory mechanism of grain hardness. However, I think this manuscript still has the following problems:

1. Whether the periods of the four varieties tested by proteomics are consistent? Please add clarification.

2. In this manuscript, only biogenic analysis was carried out, and transcriptional expression pattern verification was lacking.

3. The experiments in the part of materials and methods are not reflected in this manuscript. Please check them carefully.

4. Whether the pathway chosen in KEGG is a significant enrichment pathway, please explain in this paper.

5. It is suggested that the authors supplement the phenotypic maps of the four wheat varieties mentioned in the paper.

6. The breed name is misstated in the sentence ---“Of these, 67 proteins were significantly up-regulated in CM605 compared to MY26, CX3, and ZM1, and were classified in cluster 4 ”. Please check it carefully.

7. “Triticum aestivum L.” Should be changed to  Triticum aestivum L.

8. Authors should try to cite references from the last 5 years.

9. The font size of the picture in Fig3 should be consistent.

10. The legend description of Fig4 should be supplemented with details, such as what bubble size represents.

Reviewer 4 Report

I enjoyed reading the manuscript. Results are interesting.

Few comments:

1. Can authors show one of the proteomic data by using 2-D gel electrophoresis? 

2. There is no gene expression data for the identified proteins. I want to see the correlation data between protein amount and transcript level. Also I am curious to see the modulation of the prootein level by post-translational modification. Can authors show the expression patterns of the genes encoding the proteins upregulated or down regulated in hard or soft grains? Do not have to test all. Need to try some of the proteins.

If these data are added in the revised manuscript, data will be more believable.
